# Exercise and Occupational Stress among Firefighters

**DOI:** 10.3390/ijerph19094986

**Published:** 2022-04-20

**Authors:** Elpidoforos S. Soteriades, Paris Vogazianos, Federica Tozzi, Athos Antoniades, Eleftheria C. Economidou, Lilia Psalta, George Spanoudis

**Affiliations:** 1Healthcare Management Program, School of Economics and Management, Open University of Cyprus, Nicosia 2220, Cyprus; 2Environmental and Occupational Medicine and Epidemiology (EOME), Department of Environmental Health, Harvard T.H. Chan School of Public Health, Boston, MA 02115, USA; 3Department of Social and Behavioral Sciences, School of Humanities, Social and Education Sciences, European University Cyprus, Nicosia 1516, Cyprus; paris.vogazianos@stremble.com; 4Stremble Ventures Ltd., Limassol 4042, Cyprus; federica.tozzi@stremble.com (F.T.); athos.antoniades@stremble.com (A.A.); 5Faculty of Medicine, School of Health Sciences, University of Ioannina, 45110 Ioannina, Greece; eleftheria.economidou@gmail.com; 6School of Sciences, UCLan Cyprus, Larnaka 7080, Cyprus; liliapsalta@gmail.com; 7Department of Psychology, University of Cyprus, Nicosia 1678, Cyprus; spanoud@ucy.ac.cy

**Keywords:** firefighters, exercise, occupational stress, COPSOQ, DASS survey, Cyprus

## Abstract

The aim of our study was to evaluate the potential association between physical activity and occupational stress among firefighters. Data were collected from Cypriot firefighters through a web-based battery of internationally validated questionnaires completed anonymously (COPSOQ, DASS). A total of 430 firefighters (response rate 68%) completed the survey (age range: 21–60 years). More than half of the firefighters (54%) reported either no or minimal physical activity. A total of 11% of firefighters reported moderate to extremely severe stress based on the DASS-S scale. Using multivariable-adjusted logistic regression models, we showed that firefighters who exercised had 50% lower risk of occupational stress, and using a categorical model, we found that every hour per week of increased physical activity among firefighters was associated with 16% lower risk of occupational stress after adjusting for age, education, smoking, and body mass index (OR = 1.16; *p* = 0.05). In addition, our findings suggest an inverse dose–response relationship between physical activity and occupational stress among firefighters. Physical activity appears to be inversely associated with occupational stress and serves as an important mitigating factor of occupational stress in firefighters. Further research is warranted to evaluate the potential effect of exercise interventions on occupational stress, and the overall mental health of firefighters and other occupational groups.

## 1. Introduction

Regular exercise is associated with several positive physical and psychological health benefits. In numerous studies, researchers reported several positive health outcomes associated with exercise, including relatively strong association with psychological wellbeing in the general population [1,2]. Increased participation in exercise is also associated with lower risk of cardiovascular morbidity and overall mortality [3]. Particularly among firefighters, regular exercise (resistance and aerobic) is extremely important and is associated with many positive health outcomes, including improved aerobic capacity, lower body fat, and increased strength, endurance, and power [4]. Furthermore, firefighters’ higher exercise levels were associated with lower cardiovascular-disease risk factors [5,6], while exercise contributes to better respiratory performance [7] and improved endurance when wearing a self-contained breathing apparatus (SCBA) [8]. In addition, the fittest firefighters had the lowest risk of on-duty injuries [9].

Levels of exercise vary significantly across different populations and occupational groups [10]. In particular, several studies reported firefighters having clusters of cardiovascular disease risk factors and exhibiting low levels of cardiorespiratory fitness. Specifically, Risavi et al. concluded that many Pennsylvania firefighters (USA) had risk factors for heart disease (14% were smokers, 41% had HTN, 38% had pre-HTN, with only 12% receiving treatment, and 13.5% were treated for high cholesterol) [11]. Moreover, in a study that examined the prevalence of metabolic syndrome (MetS) in association with cardiorespiratory fitness levels in Colorado firefighters, about 1 in 10 had MetS, and nearly half of the sample had insufficient cardiorespiratory fitness [12]. Another study found that firefighters had similar or better cardiometabolic health profiles than those of the US general population; however, both males and females had a high prevalence of cardiovascular risk factors, which continued to worsen with increasing age [13]. Therefore, many researchers argue that there is an urgent need for firefighters across the globe to improve their levels of exercise and physical fitness in order to protect their health, and reduce their risk of injuries and fatalities [14,15]. No studies have examined the potential association of exercise with mental health among firefighters, particularly with occupational stress.

According to the WHO Network of Collaborating Centers in Occupational Health, stress that is developed in or exacerbated by work in considered to be work-related (occupational) stress. Occupational stress constitutes one of the most globally prevalent work-related health problems [16]. This is described as the adverse reaction experienced by workers when workplace demands and responsibilities are greater than those that the worker can comfortably manage or are beyond the workers’ capabilities. According to the WHO, stress-related illness constitutes the greatest cause of early deaths in European countries, and work-related stress is associated with a high cost to employees, employers, and society in general [17].

Occupational stress affects employees from many different workplace settings and industries. Particularly, employees who provide services to people and interact with clients are more prone to the development of occupational stress, including teachers, physicians, police officers, and firefighters [18,19,20]. Firefighters constitute an occupational group with relatively high levels of stress associated with their job duties. Many reasons may contribute to this observation. Firefighters are directly exposed to harmful agents, stressful situations, and different hazards at work that could lead to the development of acute and/or chronic occupational stress throughout their career [21]. Firefighters’ duties, such as rescuing victims from motor vehicle accidents or fire scenes, may also be associated with extreme mental stress, particularly when involving the fatalities of children and young adults [22]. Firefighters directly deal with people’s lives, which requires full awareness and swift life-or-death decision making [21]. Such events may lead to the development of post-traumatic stress disorder and chronic occupational stress [23,24]. Moreover, occupational stress has psychological (anxiety, depression, irritability, aggression, the inability to reach decisions, poor concentration, distractions), physical (migraine headaches, hypertension, cardiovascular pulmonary and kidney disease, musculoskeletal sleep and immune system disorders, rheumatoid arthritis), and organizational effects (absence from work, turnover, low production, job dissatisfaction, reduced organizational commitment, decline in occupational performance) on employees and organizations [21]. Additionally, prolonged shift work and poor communication with co-workers and managers (poor employee interpersonal relationships) may also serve as strong stressogenic workplace factors [25,26,27].

The beneficial effects of exercise on occupational stress were examined among different occupational groups [28]. Despite firefighting being one of the most stressful occupations, research on the particular association between exercise and work-related stress among firefighters has not been reported. Furthermore, although regular exercise appears to be a strong predictor of psychological wellbeing and a promising intervention to mitigate the effects of occupational stress in the workplace, no reported workplace interventions have examined the above potential effect among firefighters.

The objective of our study was to examine the levels of exercise among firefighters, and explore the potential association between exercise and occupational stress in firefighters.

## 2. Methods

### 2.1. Study Participants

Our cross-sectional study was conducted in the Cyprus Fire Service, which constitutes the public organization of professional firefighters in Cyprus. All professional firefighters from all districts of the island and all regional fire stations in Cyprus (a total of 33 fire stations) were invited to participate in a web-based survey. The study was approved by the Cyprus National Bioethics Committee, and the confidentiality of each participant was protected through a web-based application that was available on the official website of the Cyprus Fire Service. All participants gave informed consent before answering the online questionnaire.

### 2.2. Data Collection

Data were collected through anonymously completed questionnaires that were translated into the Greek language using the back-to-back translation method with qualified translators. This procedure involved a forward translation from the original language (English) to the intended language (Greek) and was followed by a back translation of the intended language (Greek) into the original language (English) and a comparison with the original version. Inaccuracies in the intended language were simply identified through differences in meaning that occurred in the backward translation. 

To further prevent any inaccuracies from the original version due to translation, we pilot-tested the battery of assessment tools in a random group of 30 firefighters. The pilot sample was randomly selected from the entire firefighter population. The aim of the pilot study was to: (i) determine whether the participants understood the questions within their cultural boundaries, (ii) assess the feasibility of the full-scale survey that followed, and (iii) establish the effectiveness of the sampling frame and used techniques. The minor adaptation of survey items also took place following this pilot study. The questionnaires included open-ended questions, Likert scale questions, and binary response items. Questionnaires were completed individually, and all responses were stored in a secure electronic database. The web-based survey was conducted in 2015 and was available to all firefighters of the Cyprus Fire Service.

Demographic information was collected on all study participants and included categorical age, gender, marital status, education, job position or ranking, smoking, body mass index (BMI), and overall health status and wellbeing. Specific inquiries also included questions on the engagement of firefighters with exercise. Psychosocial working conditions, and health and wellbeing were evaluated using the Copenhagen Psychosocial Questionnaire (COPSOQ) [29]. Firefighters’ depression, anxiety, and stress were assessed using the Depression, Anxiety and Stress Scale (DASS) [30]. For the purposes of our survey, we used the short version of the DASS scale (DASS-21), which consists of 21 items, and all questions are evaluated on a Likert scale. In principle, a 4-point (ranging from 0 to 3) severity scale measures the extent to which each state had been experienced over the past week, with 0 representing “Did not apply to me at all”, and 3 “Applied to me very much or most of the time”. The DASS-21 scale is further subdivided into 3 subscales, namely, depression, anxiety, and stress, using 7 items for each subcategory.

### 2.3. Explanatory and Outcome Variables

Exercise was reported by firefighters as the number of days per week that they participated in some activity and the time engaged in exercise per day. The final outcome variable was calculated as the average days per week that a firefighter would report exercising multiplied by the average time that firefighters would exercise per day. This outcome was converted into a number of hours of exercise per week by dividing the overall outcome by 60. Lastly, the distribution of exercise among firefighters was categorized into 6 groups. The first group included firefighters who reported zero exercise per week, the second group included those who were exercising up to 1 h per week (0 to 1 h), the third group included firefighters exercising between 1 and 2 h per week, the fourth group those who were exercising between 2 and 3 h per week, the fifth group included firefighters who exercised between 3 and 4 h per week, and the final group included firefighters exercising more than 4 h per week. The categorization aimed at exploring time of exercise in hours per week, and the final category was constructed to correspond to the recommendations of the international guidelines for regular exercise [31].

Following the appropriate guidelines to calculate the DASS stress score, we first constructed the overall score by adding the DASS stress components (DASS_1 + DASS_6 + DASS_8 + DASS_11 + DASS_12 + DASS_14 + DASS_18). This led to the construction of a DASS stress variable that extended from a minimal value of zero to a maximal score of 21. The DASS stress score was then transformed by multiplying the outcome by a factor of 2 in order to bring the minimal value to 0 and the maximal value to 42. Following this, we separated our sample into 5 ordinal groups according to their final score as follows: normal (0–14), mild stress (15–18), moderate stress (19–25), severe stress (26–33), and extremely severe stress (34+). In addition, we constructed a dichotomous variable of stress among firefighters, with the reference category being those firefighters without stress (DASS stress score: 0–14), and the comparison group included firefighters with stress (DASS stress score: 15+) [32]. 

### 2.4. Statistical Analyses

Data were imported into the XLStat statistical package for statistical analyses. Cross-tabulations were performed using SPSS v23 (IBM, New York, NY, USA), while regressions were run by XLSTAT (Microsoft, Redmond, WA, USA). Contingency table analysis was performed to examine the association of population demographics with the outcome of occupational stress. Joint frequency distribution was analyzed using Pearson’s chi-squared test to determine whether the different demographic variables were statistically independent or whether they were associated with the outcome of interest.

Bi- and multivariable-adjusted logistic regression models were used to assess the potential association between exercise and occupational stress independent of other covariables [33,34]. For odds ratios below 1 (protective effect), we used a reciprocal approach by dividing them by 1 in order to facilitate the average reader interpreting the results. Exercise was the main explanatory variable in our regression models and it was used as a dichotomous variable (yes/no for reported exercise by firefighters) or as a categorical variable divided into 6 categories as described above. Occupational stress was used as a binary outcome (dependent variable), dichotomized as described above. Adjusted variables were used into the multivariable adjusted models as dichotomous variables (age above or below 35, education up to high school versus higher educational level, smoking (yes/no), and body mass index (BMI) below or above 25). Smoking and body mass index were included into the multivariable-adjusted models on the basis of their reported association with stress. Educational level was included in the multivariable model because it was statistically significantly associated with the outcome of interest (occupational stress) in our dataset on the basis of the findings in Table 1. The significance level was set at *p* = 0.05 and was two-sided for all tests.

## 3. Results

A total of 430 firefighters (response rate 68% among all professional firefighters) completed the survey questionnaires (380 males (88.4%), and 50 females (11.6%)). Firefighter age ranged from 21 to 60 years old. The distribution of age categories among participating firefighters was the following: 18.1% (21–30 years of age), 38.7% (31–40 years of age), 26.7% (41–50 years old), and 16.5% for 51–60 years old. A total of 10.7% reported being occasional smokers, and 32.1% were regular smokers. In addition, 45.6% of firefighters were overweight, and 12.5% were obese. The majority of the study participants (76.6%) were ranked as firefighters, 14.8% were sergeants, and 8.6% were senior officers. 

In Table 1, we present the demographic characteristics of firefighters by occupational stress categories. About 16.6% of firefighters reported no exercise at all, while 37.5% engaged in some form of exercise 1–2 times per week, one-third of firefighters (30.8%) reported exercising 3–4 times per week, 8.6% reported exercising 5–6 times per week, and 6.5% reported daily exercise. The distribution of exercise duration (hours of exercise per week) as used in the logistic regression models was the following: up to 1 hour of exercise per week (19.5%), 1 to 2 h of exercise per week (26.9%), 2 to 3 h per week (14.7%), 3 to 4 h per week (8.2%), and more than 4 h per week (14.2%). About one-third of firefighters were regular smokers, and 1 in every 8 firefighters were obese. Educational level was inversely associated with occupational stress (*p* = 0.015). In addition, there was a trend towards significance when examining marital status and occupational stress. Single and divorced or widowed firefighters were more likely to report stress compared to married firefighters.

The internal consistency estimates of the reliability of the scales used were quite high, as expressed with Cronbach’s α, which was 0.91 for the DASS stress scale. Notably, the correlation between the DASS stress and DASS depression subscales was 0.85 and statistically significant (*p* < 0.001). Using the DASS stress scale, we found that 83.3%, 5.5%, 7.7%, 3.1%, and 0.5% of the sample were categorized into the normal, mild, moderate, severe, and extremely severe subcategories of stress, respectively. In Table 2, we present bivariable and multivariable adjusted logistic regression models for the inverse association between exercise and occupational stress in firefighters.

Multivariable logistic regression models were adjusted for age, education, smoking, and body mass index. Using the dichotomous exercise variable, we found that firefighters who reported any exercise had a 50% lower chance of having occupational stress compared to those who did not exercise at all. The same effect was also retained in the multivariable adjusted model although it did not reach statistical significance. In further examining the association of a categorical variable of exercise as presented in Table 1 (hours of exercise per week), we found that for every additional hour of exercise, firefighters were 16% less likely to report occupational stress and the effect was statistically significant and was also retained in the multivariable adjusted model. Furthermore, we explored a potential dose–response relationship between exercise and occupational stress in simple and multivariable adjusted regression models. However, a dose–response relationship was not evident due to relatively small numbers and lack of statistical power. Firefighters who reported exercising up to one hour per week were 50% less likely to report occupational stress, while firefighters who were exercising between 1 to 2 h per week were 2 times less likely to report occupational stress compared to those who did not exercise.

## 4. Discussion

To the best of our knowledge, this is the first study to evaluate the association between exercise and occupational stress among firefighters. A troublesome finding of our study was that about 1 in 6 firefighters reported no exercise at all, while an additional one-third of firefighters reported exercising only 1–2 times per week (Table 1). In total, more than half of firefighters either did not exercise at all or were involved in exercise at a minimal level. These findings are in line with other published studies reporting on the disturbingly low levels of exercise and cardiorespiratory fitness among firefighters. Li et al. reported that, among Colorado firefighters, only half of both males and females met the minimal cardiorespiratory fitness level [12]. Similarly, Achmat et al. reported in another study that more than half of South African firefighters were not participating in regular physical activity [35]. Noteworthy is the conclusion of Baur et al. that the total weekly duration of aerobic exercise and of weight lifting sessions in male career firefighters was declining with aging [36].

In our study, more than 11% of firefighters were categorized into the moderate, severe, or extremely severe stress category using the DASS stress scale. The most important of our findings was the statistically significant inverse association between exercise and occupational stress. Firefighters who reported exercising were less likely to report occupational stress. In particular, for every additional hour of exercise, firefighters were 16% less likely to report occupational stress (Table 2). Furthermore, firefighters who engaged in exercise more than 4 h per week were 30% less likely to report occupational stress (Table 2).

A comparison of our findings with previous studies reveals that the vast majority of international reports show similar inverse associations between exercise and lower stress levels in the general population [37,38,39], and among different populations of employees [28,40,41]. Furthermore, a number of studies reported similar findings among employee categories that are considered to be closer to firefighters, such as the military, where lower levels of leisure-time physical activity were associated with psychological distress [42,43]. In light of the evidence documented among firefighters from Cyprus, the association under study merits particular attention in order to further explore this finding among firefighters. Given the relatively high levels of work-related stress among this occupational group, the potential beneficial effects of health promotion and interventional exercise programs constitute a promising field.

In our study, educational level was inversely associated with occupational stress. It appears that educated firefighters exhibit higher motivation for coping with stress. On the basis of the results of a study by Fardin, stress management education is effective in keeping the stress score constant and preventing it from worsening [44]. In addition, in our study, single and divorced or widowed firefighters were more likely to report stress compared to married firefighters. Sepidarkish et al. found that single participants had a high level of stress. Dividing home-based duties with a companion and enjoying emotional support could be considered as protective factors against stress [45].

The strengths of our study include the relatively high response rate in the survey of the firefighters, which minimizes selection bias. In addition, we used a widely validated questionnaire for evaluating our explanatory variables and outcome of interest, and employed several different exposure categorizations along with age- and multivariable adjusted regression models to evaluate the association of interest. 

Limitations include not considering the type and intensity of the performed exercise. Moreover, we did not use any explanations to the participants to help them in understanding what they could count as exercise. Additional limitations include the relatively small number of participants and the cross-sectional study design, limiting our ability to explore causal associations and directionality [46,47]. In addition, we used a subjective measure of exercise via survey data collection, which is likely to introduce information bias. Supporting evidence regarding the association between exercise and occupational stress in the current study requires further assessment using a prospective study design and/or a workplace intervention in order to explore the possible underlying causal relationships and evaluate the effect of exercise interventions on firefighters’ mental health.

## 5. Conclusions

Considering the current findings, more than half of firefighters either do not exercise or they engage in minimal exercise. In parallel, a relatively high percentage of firefighters reported occupational stress. More importantly, a statistically significant inverse association between exercise and occupational stress in firefighters, even after adjusting for several other explanatory variables, was found. These study results require further investigation using a prospective study design and/or a randomized control trial in order to verify the above findings and explore potential causal relationships. Such an etiological relationship could prove to be extremely instrumental in developing exercise interventions in the fire service to improve firefighters’ mental health and overall job performance. In addition, it may have parallel important implications for other occupational groups such as paramedics, police officers, and the military.

## Figures and Tables

**Table 1 ijerph-19-04986-t001:** Demographic characteristics of the study sample by stress categories (dichotomous) (N = 430).

Characteristic	Stress Score		*p*-Value
	0–14 N (%)	15+ N (%)	
**Total**	317 (75.7)	102 (24.3)	
**Age (years)**			
21–30	59 (18.6)	17 (16.7)	
31–40	121 (38.2)	41 (40.2)	
41–50	85 (26.8)	27 (26.5)	
51–60	52 (16.4)	17 (16.7)	0.134
**Gender**			
Male	279 (88.0)	92 (90.2)	
Female	38 (12.0)	10 (9.8)	0.597
**Marital status**			
Single	46 (14.5)	16 (15.7)	
Married	247 (77.9)	76 (74.5)	
Divorced/separated/widowed	24 (7.6)	10 (9.8)	0.093
**Education**			
Primary School	0 (0)	1 (1.0)	
High School	234 (73.8)	85 (83.3)	
Diploma/bachelor’s degree	73 (23.0)	11 (10.8)	
Master’s degree or higher	10 (3.2)	5 (4.9)	0.015
**Ranking**			
Firefighters	240 (75.7)	81 (79.4)	
Sergeants	51 (16.1)	11 (10.8)	
Officer	19 (6.0)	7 (6.9)	
Other senior managers	7 (2.2)	3 (2.9)	0.475
**Smoking**			
Nonsmoker	117 (36.9)	40 (39.2)	
Ex-smoker	63 (19.9)	15 (14.7)	
Occasional smoker	33 (10.4)	13 (12.7)	
Regular smoker	104 (32.8)	34 (33.3)	0.658
**Body mass index (BMI)**			
≤25	131 (41.3)	34 (33.3)	
25–30	148 (46.7)	48 (47.1)	
≥30	37 (11.7)	17 (16.7)	0.255
**Days of exercise/week**			
Never	48 (15.1)	21 (20.6)	
1–2 times	120 (37.9)	36 (35.3)	
3–4 times	99 (31.2)	29 (28.4)	
5–6 times	28 (8.8)	8 (7.8)	
Daily	21 (6.6)	6 (5.9)	0.759
**Hours of exercise/week**			
Zero	48 (15.2)	21 (21.0)	
1 h/week	52 (16.5)	29 (29.0)	
2 h/week	94 (29.7)	18 (18.0)	
3 h/week	48 (15.2)	13 (13.0)	
4 h/week	27 (8.5)	7 (7.0)	
>4 h/week	47 (14.9)	12 (12.0)	0.029

**Table 2 ijerph-19-04986-t002:** Logistic regression models for the inverse association between exercise and occupational stress in firefighters (N = 430) *.

Logistic Regression Models	OR (95% CI)	*p*-Value
**Model 1 (exercise as a dichotomous variable, yes/no)**		
Dichotomous	1.52 (0.87–2.67)	0.14
Dichotomous (age-adjusted)	1.52 (0.87–2.67)	0.14
Dichotomous (multivariable adjusted) †	1.51 (0.86–2.66)	0.15
**Model 2 (exercise as a categorical variable, hours per week)**		
Categorical	1.17 (1.008–1.35)	0.039
Categorical (age adjusted)	1.17 (1.009–1.35)	0.038
Categorical (multivariable adjusted) †	1.16 (1.000–1.34)	0.05
**Model 3 (dose–response effect of different levels of exercise)**		
Exercise (zero hours per week)	-	-
Exercise (up to 1 h per week)	1.52 (0.87–2.67)	0.14
Exercise (1 to 2 h per week)	2.18 (1.38–3.44)	0.0008
Exercise (2 to 3 h per week)	1.35 (0.84–2.17)	0.22
Exercise (3 to 4 h per week)	1.31 (0.75–2.31)	0.34
Exercise (>4 h per week)	1.29 (0.65–2.54)	0.46
**Model 4 (multivariable adjusted dose-response effects) †**		
Exercise (zero hours per week)	-	-
Exercise (up to 1 h per week)	1.54 (0.87–2.73)	0.13
Exercise (1 to 2 h per week)	2.16 (1.36–3.43)	0.0011
Exercise (2 to 3 h per week)	1.29 (0.79–2.10)	0.30
Exercise (3 to 4 h per week)	1.24 (0.70–2.20)	0.45
Exercise (>4 h per week)	1.29 (0.65–2.56)	0.47

* Occupational stress was assessed using DASS stress subscale. † Multivariable logistic regression models were adjusted for age, educational level, smoking and body mass index.

## Data Availability

Data are available upon request.

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
