# Peer review of "Exercise and Occupational Stress among Firefighters"

_ijerph, 2022, doi:10.3390/ijerph19094986_

Round 1

Reviewer 1 Report

The study described in this manuscript is designed to assess the relationship between physical activity and stress levels among firefighters. The overall results of the study show that exercise can reduce firefighters' stress. The study was planned and conducted correctly. Although, the subject of the study is not very interesting and I doubt about novelty of the study. Because studies have already been done on the effectiveness of health intervention programs, including physical activity, on the mental health of firefighters. On the other hand, doing physical exercise and maintaining physical fitness is one of the main tasks and the nature of the firefighters' job.

Introduction section

-This section is well written but it is recommended to use more relevant and recent references.

- Lines 48 and 49: This sentence is a bit strange so it needs to be checked with other references. In pre-employment medical examinations, the aerobic and cardiovascular capacity of firefighters is usually assessed and if they meet the set values, the person will be allowed to be employed in that job. During the employment period, firefighters are required to maintain physical fitness, and their cardiovascular and aerobic status is monitored annually. Therefore, it is expected that the average cardiovascular and aerobic capacity of firefighters to be higher than others.

- Paragraph 3 (lines 63 to 76): Explain more about firefighters' occupational stressors using new and related references.

Methods section

 - Determine the type of study.

- Sampling method should be described.

- The translation method of the questionnaires is mentioned, but no explanation is provided about the method of assessing the validity and reliability of the questionnaire.

Results section

-Provide a result related to the psychometric properties of the questionnaires.

Discussion section

-Compare the results of firefighters' stress with similar studies in firefighters and other occupations.

Reviewer 2 Report

Overall Comments:

In the present study, the authors assessed occupational stressors experienced among regional firefighters via a validated online survey. The findings demonstrated a significant inverse relationship between exercise frequency/duration and psychological stress. I find the paper to be well-written and the methodology and statistical analysis/interpretation to be sound. My edits overall are minor.

Minor Comments:

Line 70: The introduction overall is written well, but this sentence needs to be re-worded. My suggestion is “Besides, firefighters’ duties such as rescuing victims from motor vehicle accidents or fire scenes may be associated with extreme…”

I appreciate the effort this research team went through for data collection, specifically the back-back translation section.

Methods and results are sound and well written for the reader.

Discussion: The research team should elaborate on (e.g. add a paragraph) specifically on education level/stress and the trend observed for marital status. I found this highly intriguing and important when considering FF health. Very much appears that a support system in place for these FF can sig. alleviate some of the occupational stressors they experience and having the education to establish that system is the first important step. I think 3-5 sentences on this would improve the paper further.

Through the discussion, I think it would improve the paper to reference your tables.

Example: Line 247, reference your table following “occupational stress (see Table XX).”

Reviewer 3 Report

  • VERY important topic!
  • Well-sourced review of the literature. Easy to see how important this topic is, very convincing argument is made for supporting these firefighters.
  • Were any questions asked about the type of exercise performed?
  • Were any questions asked about the intensity of exercise performed?
  • Did the researchers use any explanations to the participants to help them understand what they could count as exercise and what they could not?
  • While this is a strong study that helps establish the need for potential exercise interventions, without having the previously mentioned information, it is a potential severe limitation that must be mentioned.

Round 2

Reviewer 1 Report

Revision is acceptable 

Reviewer 3 Report

Thank you for addressing those concerns! While I feel that this is a great first-step in establishing the need to further evaluate the influence that exercise has on the occupational stress of this important population, it will be crucial to make sure to further evaluate both the type and intensity of exercise that these individuals tend to perform (and also enjoy!). Great pilot study, look forward to seeing the next steps.